# Unlocking full and fast conversion in photocatalytic carbon dioxide reduction for applications in radio-carbonylation

Serena Monticelli[1], Alex Talbot[1], Philipp Gotico [2], Fabien Caillé[3], Olivier Loreau[1], Antonio Del Vecchio[1], Augustin Malandain[1], Antoine Sallustrau[1], Winfried Leibl [2], Ally Aukauloo[2,4], Frédéric Taran[1], Zakaria Halime [4] ✉ & Davide Audisio [1] ✉

Harvesting sunlight to drive carbon dioxide ($CO_2$) valorisation represents an ideal concept to support a sustainable and carbon-neutral economy. While the photochemical reduction of $CO_2$ to carbon monoxide (CO) has emerged as a hot research topic, the full $CO_2$-to-CO conversion remains an often-overlooked criterion that prevents a productive and direct valorisation of CO into high-value-added chemicals. Herein, we report a photocatalytic process that unlocks full and fast $CO_2$-to-CO conversion (<10 min) and its straightforward valorisation into human health related field of radiochemistry with carbon isotopes. Guided by reaction-model-based kinetic simulations to rationalize reaction optimisations, this manifold opens new opportunities for the direct access to [11]C- and [14]C-labeled pharmaceuticals from their primary isotopic sources [[11]C]$CO_2$ and [[14]C]$CO_2$.

While fuel and natural gas prices have skyrocketed worldwide and global warming has alarming impact on our society and the future of our planet, the transformation of greenhouse gases into high-value molecules has become an impelling priority[1–3]. As such, the valorisation of carbon dioxide ($CO_2$) and its conversion into synthetically valuable C1 building blocks represents a challenge tackled with extensive efforts by the scientific community[4–9]. $CO_2$ reduction to carbon monoxide (CO) is particularly appealing to support a circular and carbon-neutral economy. Among the established methodologies, electrochemical[10,11] and photochemical (for representative reviews, see refs. 12–14) strategies for $CO_2$ reduction may provide a future opportunity for sustainable industrial applications. Conceptually, harvesting sunlight to drive $CO_2$ conversion, a process often referred to as artificial photosynthesis, is highly appealing to achieve low carbon footprint. Nonetheless, gaps remain between the photoreduction of $CO_2$ into CO and its subsequent valorisation.

In general, carbonylation reactions require relatively high CO concentration, but the CO evolved from the photoreduction of $CO_2$ usually does not meet such prerequisite. Furthermore, unreacted excess of $CO_2$ might be parasitic to the subsequent CO functionalisation with various side-reactions, such as the formation of ammonium bicarbonates/carbonate by $CO_2$–amine reaction in metal-catalysed amino-carbonylations[15,16]. In this context, achieving high $CO_2$-to-CO conversion and high purity of CO (>90%) would save additional energy-intensive purification steps. Unfortunately, completeness of $CO_2$-to-CO conversion is most often a neglected parameter[17]. Attention has been focused on the improvement of catalyst performances under high excess of $CO_2$, rather than on complete exhaustive $CO_2$ conversions. There are only limited examples in the literature where this factor has been taken into account[18]. He and co-workers have shown that a moderate $CO_2$-to-CO conversion (<10%) was observed in presence of Re(bpy)(CO)$_3$Cl photocatalyst[19], while we failed to achieve >30% $CO_2$ conversion on a scale as low as 0.5 mmol[20]. A similar outcome was encountered for the electrocatalytic reduction of $CO_2$. In 2021, the groups of Cantat and Fontecave reported that $CO_2$ electro-reduction, coupled with propylene oxide carbonylating thermal

[1]Université Paris-Saclay, CEA, Service de Chimie Bio-organique et Marquage, DMTS, F-91191 Gif-sur-Yvette, France. [2]Université Paris-Saclay, CEA, CNRS, Institute for Integrative Biology of the Cell, F-91191 Gif-sur-Yvette, France. [3]Université Paris-Saclay, Inserm, CNRS, CEA, Laboratoire d'Imagerie Biomédicale Multimodale Paris-Saclay (BioMaps), F-91401 Orsay, France. [4]Université Paris-Saclay, CNRS, Institut de chimie moléculaire et des matériaux d'Orsay, F-91400, Orsay, France. ✉e-mail: zakaria.halime@universite-paris-saclay.fr; davide.audisio@cea.fr

catalysis, provided the corresponding β-butyrolactone in only 1.7% yield from $CO_2$[21]. From a societal perspective, such results rise a challenge on the future applicability of photochemical $CO_2$ reduction.

Aiming to develop effective and stoichiometric functionalisation reactions using $CO_2$, herein we report on a photocatalytic process that unlocks full and fast $CO_2$-to-CO conversion (<10 min, at room temperature, on 0.4 mmol scale). Beside the fundamental interest of the process, the transformation has utmost implications in the field of carbon isotope radiolabeling, where [14C]CO and [11C]CO remain underexploited radioactive C1 building blocks (Fig. 1). In the process, we show the concrete application of the tandem $CO_2$ reduction/carbonylation on a variety of substrates, including pharmaceutically relevant bioactive compounds. The method is suitable to all carbon isotopes (13C, 14C and 11C).

## Results and discussion

### Carbon monoxide in radiochemistry

Carbon monoxide is an excellent C1 building block in organic synthesis and carbonylation reactions are valuable tools to elaborate organic molecules[22–26]. While [12C]CO is cheap and available at an industrial scale, this is not the case for its carbon-labeled isotopologues. While [13C]CO is the primary 13C source, high isotopic purity is obtained by energy-intensive cryogenic distillation from natural CO (1.1% 13C abundance)[27]. For radioactive 14C (β⁻ emitter, half-life 5730 years) and 11C (β⁺ emitter, half-life 20.4 min), access to radiolabeled CO is challenging. [14C]CO is unstable and undergoes radiolysis[28,29], and it must be generated in or ex situ and immediately utilized. Three routes for preparing [14C]CO are described: (a) zinc-bed reduction of [14C]CO$_2$ at 385 °C[30], (b) dehydration of 14C-labeled formic acid with concentrated sulphuric acid at 70 °C[31,32] and (c) ex situ decarbonylation of acid chloride using [14C]COGen (Fig. 1a)[33]. The thermal zinc reduction has the advantage of using directly [14C]CO$_2$, (price: 1860 $/mmol) as the primary source of 14C, but it has inherent safety and practical limitations. On the other hand, use of formic acid and COgen are milder and easier to implement, but require secondary 14C-reagents that are synthetized from [14C]CO$_2$. For short-lived 11C, access to [11C]CO is mainly granted by direct reduction of [11C]CO$_2$ by zinc (400 °C) and molybdenum (850 °C) (for recent reviews on the topic, see refs. 34,35). Due to the 11C short half-life, the use of secondary formate derivatives is unsuitable, while the reduction of [11C]CO$_2$ with silyl reagents was recently reported[36–38]. Additionally, in 2017 a proof-of-concept showing the use of electrochemical reduction of [11C]CO$_2$ was reported by Gee and Long[39]. In light of the state-of-the-art, unlocking high conversion in the photocatalytic reduction of $CO_2$ would have a significant

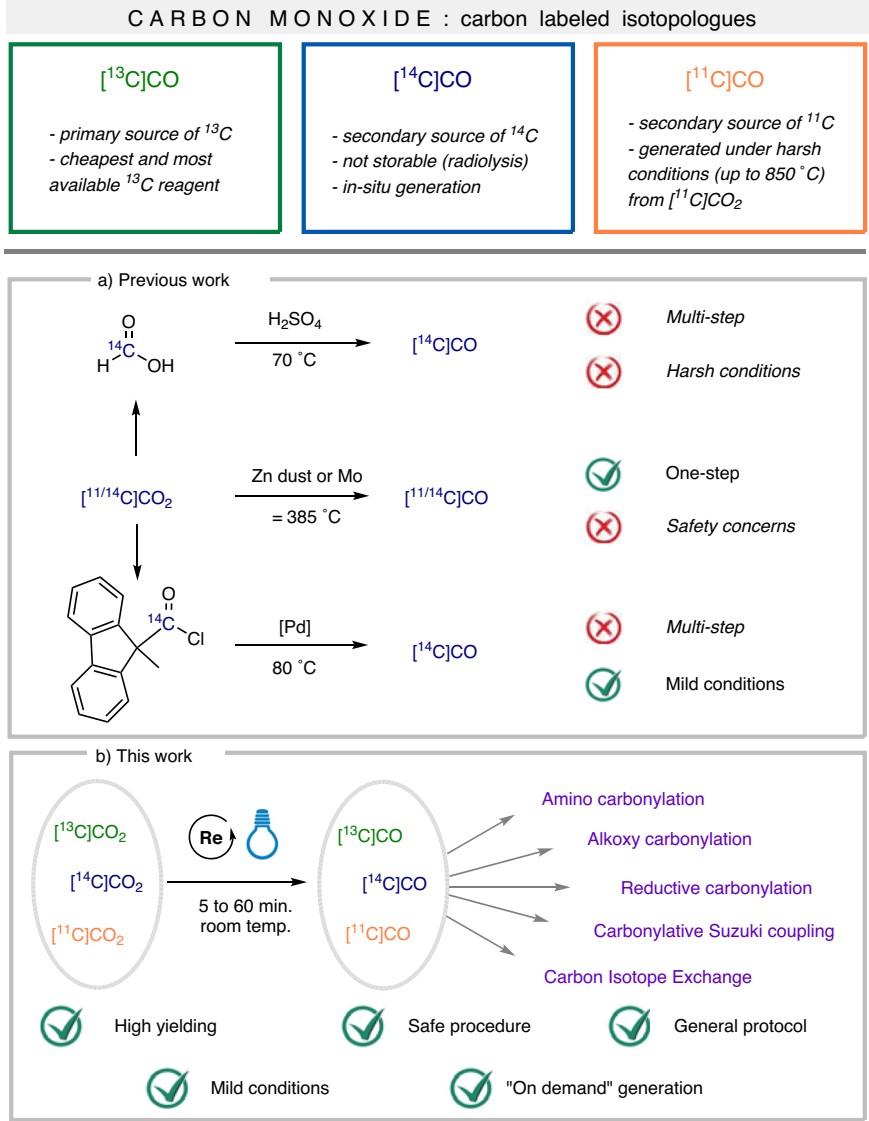

**Fig. 1 | Current state-of-the-art to access carbon isotopologues of carbon monoxide. a** Three general strategies for formal $CO_2$-to-CO reduction. **b** Our work and opportunities on the rapid and full $CO_2$-to-CO photoreduction and direct valorisation of CO.

impact on the field of $CO_2$ valorisation and major applications in carbon isotope chemistry.

Using transient absorption spectroscopy, we previously described the photo-induced electron transfer steps, from a ruthenium (II) trisbipyridine photosensitizer (**Ru PS**) to a rhenium (I) bipyridine triscarbonyl catalyst (**Re cat**)[20]. We have shown that higher efficiencies for the photocatalytic reduction of $CO_2$-to-CO were achieved using 1,3-dimethyl-2-phenylbenzimidazoline (BIH), as two-electron and one-proton sacrificial donor and water as additional proton source in dimethylformamide (DMF). A first proof-of-concept for the consecutive utilisation of the photo-produced [$^{13}$C]CO in an aminocarbonylation reaction could be obtained. However, we clearly highlighted the importance of the often-overlooked criterion of full $CO_2$-to-CO conversion for this valorisation strategy, in contrast with the commonly used criterion of turnover number (TON = amount of CO/ amount of catalyst). Indeed, we found that the lower $CO_2$-to-amide yield was due to a low $CO_2$-to-CO conversion (<30%) in the first reaction (i.e., the photocatalytic reduction of $CO_2$ by the Re catalyst). NMR monitoring has revealed that the low yield was in part due to a concomitant formation of bicarbonate during the photocatalytic production of CO, as a result of $CO_2$ acting also as an oxygen atom acceptor ($2\,CO_2 + 2\,e^- + H^+ \rightarrow CO + HCO_3^-$). To address this issue, guided by kinetic simulations (vide infra), we performed a systematic screening of different components of the catalytic system such as solvents, proton sources and additives that may play the role of oxygen atom acceptor.

## Reaction optimisation

The optimisation of the photocatalytic $CO_2$ reduction was performed using a two-chamber reactor system (Supplementary Table 1). The photoreduction took place in the first chamber (Ch.1), where a precise amount of stable [$^{13}$C]$CO_2$ was introduced with an RC-TRITEC carboxylation manifold, to guarantee a precise gas loading. The scale of labeled $CO_2$ was decided to be 0.3 and 0.4 mmol. This decision was consciously made for the application to the radioisotopes $^{14}$C and $^{11}$C for the following reasons: (a) this scale is a good compromise in terms of costs of the $^{14}$C-radioactive material; (b) it provides suitable amounts of labeled product for in vivo applications; (c) it limits the possible generation of long-lasting radioactive waste after the completion of the carbonylation reactions; (d) it avoids high pressure in the reactor and provides a more suitable safety profile for the implementation to radioactive carbon.

After the reduction and gas diffusion, the [$^{13}$C]CO generated was systematically quantified using an established palladium-catalysed aminocarbonylation reaction in the second chamber (Ch.2; yield based on [$^{13}$C]$CO_2$ as limiting reagent). Our initial reaction conditions utilizing **Re cat** (0.15 mol%), **Ru PS** (0.15 mol%) under blue light irradiation (Kessil A160WE Tuna Blue LED lamps, Supplementary Figs. 25 and 26), BIH and water as proton source allowed to obtain **[$^{13}$C]1** in a disappointing 35% yield (Fig. 2A, entry 1). Extensive optimisation pointed out the primary importance of the proton source in enhancing the efficiency of the transformation. The use of phenol revealed an improvement in comparison with the commonly used TEOA (triethanolamine) and water (Fig. 2A, entries 1–5). To our delight, by increasing the loading of **Re cat** and **Ru PS** from 0.15 mol% to 0.45 mol% (entry 8), the desired amide **[$^{13}$C]1** was obtained in 70% yield. In contrast, no beneficial effect was observed in presence of TEOA, under the same conditions (entry 4).

To prevent the collateral formation of $HCO_3^-$, triphenylphosphine (PPh$_3$, 1 equiv.) was used as a potential oxygen atom acceptor[40–43] and we were pleased to see an improvement in the yield of ca. 5–10% (entries 2–3, 6–7). Interestingly, PPh$_3$ was effective even when used in catalytic amount (10 mol%), thus disproving its role as a stoichiometric oxygen atom acceptor (entry 12). Spectro-electrochemical investigations showed no significant effect of the PPh$_3$ additive in the reduction

potentials of the **Re cat**, nor in the electrocatalytic reduction activity of the catalyst (Supplementary Figure 19). These results suggested that PPh$_3$ does not play a role in the catalytic redox cycle of **Re cat**. At present, the origin of such beneficial effect remains unclear. Additionally, we observed that the dual **Ru PS - Re cat** sensitised catalysis offered a tangible advantage in terms of reaction time, allowing the full conversion of [$^{13}$C]$CO_2$ into [$^{13}$C]CO within only 1 h (entry 10). In absence of **Ru PS** the photoreduction was much slower (after 1 h only 40% yield of **[$^{13}$C]1** is observed). Finally, by adjusting the amount of phenol in solution (5.5 equiv.) and using a catalytic amount of PPh$_3$ (10 mol%) an overall 83% yield of **[$^{13}$C]1** from [$^{13}$C]$CO_2$ was achieved (entry 12). The high isotopic purity of the compound **[$^{13}$C]1** (97.3%) highlighted that only a negligible isotopic dilution took place in the process[4]. Control experiments performed in the absence of BIH, **Re Cat** or light irradiation highlighted no product formations (entries 29, 30, 34, Supplementary Table S1). In the absence of PPh$_3$, **[$^{13}$C]1** was observed in 45% yield under identical catalyst loading, while in 55% yield when the **Ru PS–Re cat** charge was increased to 0.75 mol% (entries 32, 33, Supplementary Table S1). Experiments performed by replacing the **Re Cat** by organic photocatalyst 4CzIPN were unsuccessful (Supplementary Table 1, entries 35–37) (for a recent review on organophotoredox catalysis, see ref. 44).

Next, we screened a series of phosphines (Fig. 2B). Electron-rich phosphines (**P4-P5**) resulted to inferior outcomes while dimeric ones (**P6-P7**) even dramatically decreased the yield. In contrast, electron-poor phosphines (**P2-P3**) behaved analogously to PPh$_3$. **P2** was selected for further experiments, as a good compromise between yield and cost. We then examined the impact of substituents on the proton donor (Fig. 2C): p-methoxyphenol gave a lower yield (56%), p-nitrophenol led to no conversion (possibly be explained by interference in light absorption due to the intense yellow-colored solution); while 2,4-difluorophenol was comparable with unsubstituted phenol. A control experiment using sodium phenoxide did not show any conversion and confirmed the crucial role of phenol as a proton donor.

To evaluate in real-time the effect of reaction conditions (proton sources, solvents and light intensities) on the [$^{13}$C]CO production, we decided to undertake a series of pressure studies. With carbon monoxide being much less soluble than $CO_2$ in organic solvents, the increase in pressure measured in the headspace of the manifold directly correlates to the $CO_2$-to-CO conversion (Supplementary Fig. S10). As shown in Fig. 2D, the presence of phenol allowed reaching a plateau within only 40 min in DMF. This result shows that phenol is significantly more effective compared to other proton sources (TEOA, TFE or water, Supplementary Fig. S14). Interestingly, acetonitrile provided even better performances compared to DMF and DMSO in terms of reaction rate, as the plateau was reached within 27 min (Supplementary Fig. S15). The blue light intensity of commercially available Kessil LED lamps has been evaluated and the use of higher intensity allowed the full $CO_2$-to-CO conversion in only 7 min (Supplementary Fig. S16). Gas chromatography monitoring of the reaction headspace further confirmed these results (Supplementary Figs. S21 and 22). Interruption of the light irradiation by a series of dark/light cycles, showed that the catalyst is still active and the reduction proceeds after short pauses (Supplementary Fig. S11). Unfortunately, attempts to recycle the photocatalytic set-up by recharging the reactor with additional [$^{13}$C]$CO_2$ after the first run were unsuccessful.

These results revealed that a significant enhancement of the $CO_2$-to-CO conversion (up to 83% of **[$^{13}$C]1**, over two steps) was achieved when phenol was used as a proton source instead of water (Fig. 2D). A prior electrocatalytic evaluation of the catalytic system has also shown that for a $CO_2$-purged DMF solution of the Re catalyst, higher catalytic rates were observed when PhOH was used instead of water (Supplementary Figs. S17 and S18). Similar observations have been reported for the electrocatalytic activity in acetonitrile solutions[45].

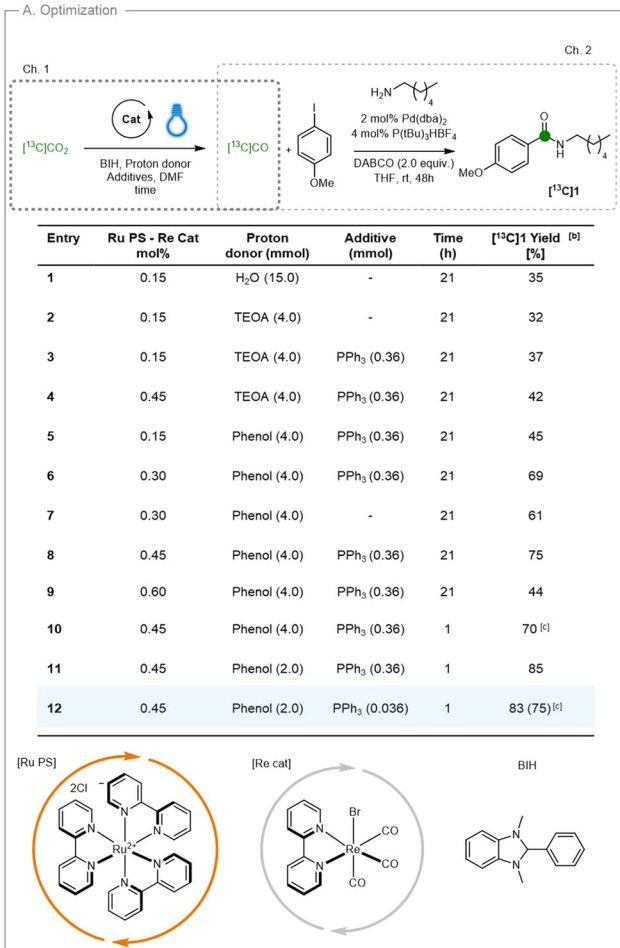

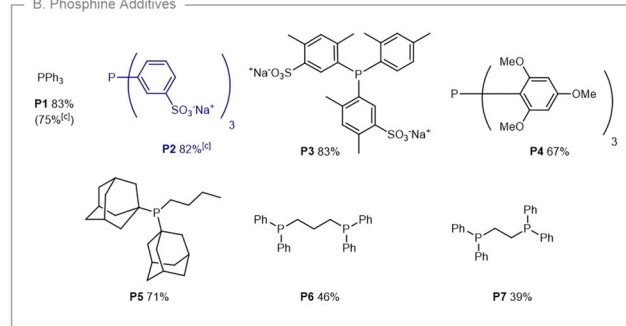

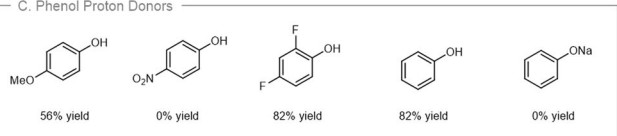

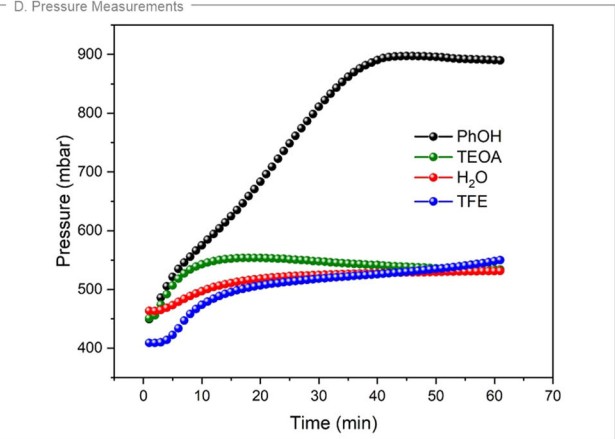

**Fig. 2 | Optimisation of the CO₂-to-CO reduction coupled with aminocarbonylation.** **A** Optimisation of the transformation. See Supplementary Tables 1 and 2, for full experimental details. [a] Ch.1: $[^{13}C]CO_2$ (0.365 mmol), BIH (0.78 mmol); DMF, room temperature; Ch.2: 4-iodoanisole (0.724 mmol), *n*-hexylamine (1.45 mmol), DABCO (1.45 mmol), Pd(dba)₂ (2 mol%), P(*t*Bu)₃HBF₄ (4 mol%), THF (0.24 M), 25 °C, 48 h. [b] ¹H-NMR yields calculated using 1,3,5-trimethoxybenzene as internal standard. [c] Yield of isolated product. DMF dimethylformamide, THF tetrahydrofuran, BIH 1,3-dimethyl-2-phenylbenzimidazoline, TEOA triethanolamine, PPh₃ triphenylphosphine, DABCO 1,4-diazabicyclo[2.2.2]octane, Pd(dba)₂ Palladium(0)

bis(dibenzylideneacetone), P(*t*Bu)₃HBF₄ Tri-*tert*-butylphosphine tetrafluoroborate. **B** Screening of phosphine additives: conditions reported in *Entry 12* (**A**) were used (0.036 mmol). **C** Screening of phenol proton donors: phosphine **P2** was used (0.036 mmol). **D** Pressure measurements relating to the rate of CO formation. Conditions: $[^{13}C]CO_2$ (0.40 mmol), BIH (0.87 mmol); proton source (2.22 mmol); **Ru PS** (0.45 mol%); **Re cat** (0.45 mol%) in DMF. The reaction mixture was stirred at room temperature under blue light irradiation (low intensity) for 1 h. TEOA triethanolamine, TFE trifluoroethanol.

However, the effect of phenol has rarely been documented to improve the activity and CO₂ conversion efficiency of photocatalytic systems involving the Re bipyridine catalysts, where most of the studies use triethanolamine, triethylamine, or water as additives in DMF or ACN[46–66]. From this optimisation, we found that the most important parameters to influence the CO₂ to CO conversion are the use of phenol as proton donor, the stoichiometry of phenol and the increase of the catalyst loading from 0.15 mol% to 0.45 mol%.

## Mechanistic studies

To build a more comprehensive picture and to understand the origin of the beneficial effect of phenol, we combined thermodynamic data (redox potentials) obtained from electrochemical experiments and DFT calculations reported in the literature, along with known p$K_a$ values, kinetic rate constants, and solubility constants of the gases (CO₂ and CO), to implement a reaction-model-based kinetic simulation (Supplementary pages 33–37). The relevance of these simulation results was experimentally validated by monitoring the pressure changes in the headspace during the photocatalytic reduction of CO₂, which gives a direct access to a time-resolved CO production profile (Fig. 3a).

Our proposed mechanism for the photocatalytic CO₂-to-CO reduction is depicted in Fig. 3d. Using transient absorption spectroscopy, we have previously shown that the catalytically active species (**Re⁻**) is generated by two successive electron transfers[20]. The first one comes from the reduced Ru photosensitizer (the formal **Ruᴵ**), to form the singly-reduced Re (**Re**) catalyst (Reaction 3 in Fig. 3d), and the second one from the highly reducing BI• radical (Reaction 4) formed during the photo-induced first steps (Reactions B and C)[67]. The next steps (Reactions 5–11) are mainly based on reported DFT calculations[68,69]. In Reaction 5, the catalytically active species **Re⁻** is nucleophilic enough to react with CO₂ and form a **ReCO₂⁻** intermediate which undergoes a first protonation (Reaction 6) to give **ReCO₂H** followed by an electron transfer step (Reaction 7) to yield **ReCO₂H⁻**. These last two steps are considered to be fast and precede a 'dehydroxylation' reaction, which is the rate-limiting step of the catalytic system. A proton can 'dehydroxylate' the **ReCO₂H⁻** intermediate to form water (Reaction 8), or another CO₂ substrate can 'dehydroxylate' this intermediate to form bicarbonate (Reaction 9). We believe that the latter is the critical step for the CO₂-to-CO conversion efficiency. Indeed, in the presence of water, the carbonic acid (H₂CO₃) produced by the equilibrium between water and CO₂ (Reaction 10) can be a

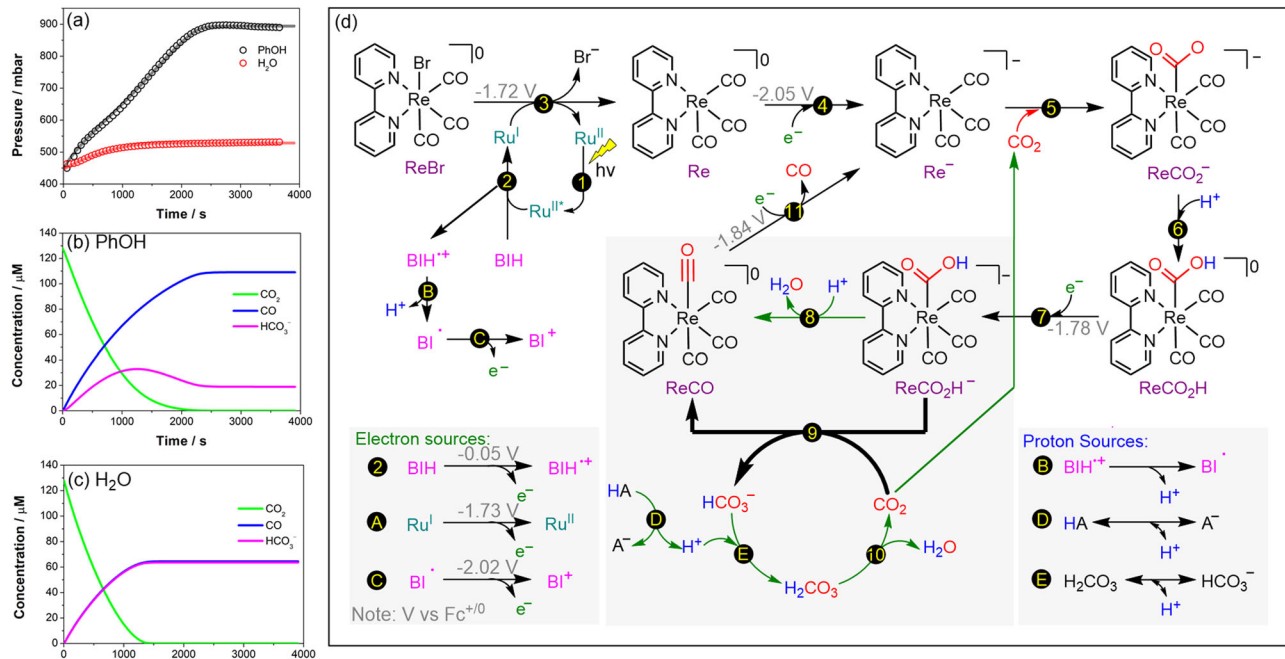

**Fig. 3 | Mechanistic investigation. a** Experimental pressure reading (circles) during the photocatalytic reduction of $CO_2$ to CO from Ch.1 containing 0.57 mM **Ru PS**, 0.57 mM **Re cat**, 279 mM BIH, 12.7 mM sodium phosphinidynetris(benzene sulfonate) and 712 mM phenol (or $H_2O$) in dimethylformamide, irradiated using a Blue LED lamp (irradiance of 117 W m$^{-2}$). Fitted simulated pressure data (solid line) is shown based on a reaction-model-based kinetic simulation. Concentration profiles of CO, $CO_2$, and $HCO_3^-$ are shown based on the results of the kinetics simulation (see Supplementary pages 28–38) distinguishing the effect of adding **b** phenol or **c** water to the photocatalytic solution. **d** Proposed photocatalytic cycle for the Ru-Re bimolecular system on which the kinetic simulation is based. Effect of water as proton source is indicated in thick black arrows accumulating bicarbonate while the changes when phenol is used is indicated in green arrows.

source of proton (consequently producing bicarbonate as shown in Reaction E), and together with Reaction 9 can generate another equivalent of $HCO_3^-$. This explains the low $CO_2$-to-CO conversion observed initially in presence of water because, virtually, 50% of $CO_2$ is converted to $HCO_3^-$ (Fig. 3c). By using phenol as a proton source in the catalytic system, the $HCO_3^-$ can be recycled back to $H_2CO_3$ (Reaction E) then to $CO_2$ (Reaction 10). It is worth mentioning that both the simulation and the pressure monitoring experiment show two distinct kinetic processes in presence of phenol, i.e., the first 20 min where the $CO_2$ originally introduced in the reaction is consumed and the second phase where the $CO_2$ recycled from bicarbonate is consumed (Fig. 3b). Phenol also plays a direct role as a proton donor in Reactions 8 and D, as further shown in Supplementary Fig. S24. In the absence of phenol, the concentration of the $HCO_3^-$ formed in the first phase remains unchanged (Fig. 3c). In the last step (Reaction 11), one more electron transfer (possibly coming from Reaction C) is needed to release the CO and regenerate the active species **Re⁻**.

### Substrate scope

Next, we attempted to apply this technology for the direct preparation of high-value ¹³C-isotopically labeled compounds exploiting the effective ex situ generation of [¹³C]CO (Fig. 4). Aminocarbonylation gives access to a straightforward synthesis of amides by using a substoichiometric amount of carbon monoxide (Fig. 4A). Combinations of (hetero)aryl iodides and alkyl amines allowed the preparation of amides containing relevant motifs, such as morpholine, piperidine and more sterically hindered adamantyl in good yields from [¹³C]$CO_2$ (compounds **[¹³C]1–6**). When higher stoichiometry of [¹³C]$CO_2$ was used (0.6 mmol), the isolated yield of amide **[¹³C]6** increased from 53% to 65%. A visible-light-enabled aminocarbonylation of alkyl iodide was attempted, as well. This non-optimised result allowed observing the labeled amide **[¹³C]3b** in 18% yield in the reaction crude. Using alcohol as a nucleophilic partner, a series of ¹³C-labeled esters was synthesized.

In the two-chamber system, the coupling reaction catalyzed by Pd(dba)₂ and CataCXium-A gave lower yields when aryl bromides were used (compounds **[¹³C]7-9**, 27–35%)[70]. On the other hand, with (hetero) aryl iodides improved yields were observed and compounds **[¹³C] 10–13** could be isolated in 64 to 75% yields.

As aldehydes are versatile functional groups in organic synthesis and found even as active pharmaceutical ingredients[71], we explored whether this technology could be used to access them through a reductive carbonylation. By applying a reported protocol in combination with our photoreduction, it was possible to obtain benzaldehydes **14–19** (Fig. 4C)[72]. Notwithstanding the volatility issue encountered for some products, we were able to prepare a set of aldehydes using [¹³C]CO as limiting reagent (compounds **[¹³C]14–16**). For compounds **[¹³C]17–19**, excess of [¹³C]CO was utilized in order to obtain satisfactory yields. Carbonylative Suzuki−Miyaura coupling was also explored for preparing nonsymmetrical benzophenones (Fig. 4D)[73]. Compounds **[¹³C]20–25** were isolated in 34 to 91% yields applying slightly modified conditions with respect to the literature. In particular, the presence of PPh₃ favored the carbonylative process, thus reducing classical non-carbonylative Suzuki−Miyaura coupling. Once more, with higher stoichiometry of [¹³C]$CO_2$ (0.6 mmol), the isolated yield of **[¹³C]21** increased from 34% to 88%. Interestingly, the presence of aryl chlorides was tolerated under different coupling conditions (see products **[¹³C]15** and **[¹³C]24**). When we attempted replacing aryl iodides with the corresponding aryl chlorides, no desired product was observed (see SI for details).

The emergence of Carbon Isotope Exchange (CIE)[74,75] provided a paradigm change in the preparation of carbon-labeled molecules[76–81]. The work by Gauthier and co-workers is particularly attractive in this area[82]. Utilizing acyl chlorides and palladium catalysis under [¹³C]CO atmosphere, the corresponding carboxylic acids were obtained without the need for time-consuming syntheses of precursors. By adapting our photocatalytic $CO_2$-to-CO conversion, we could directly access

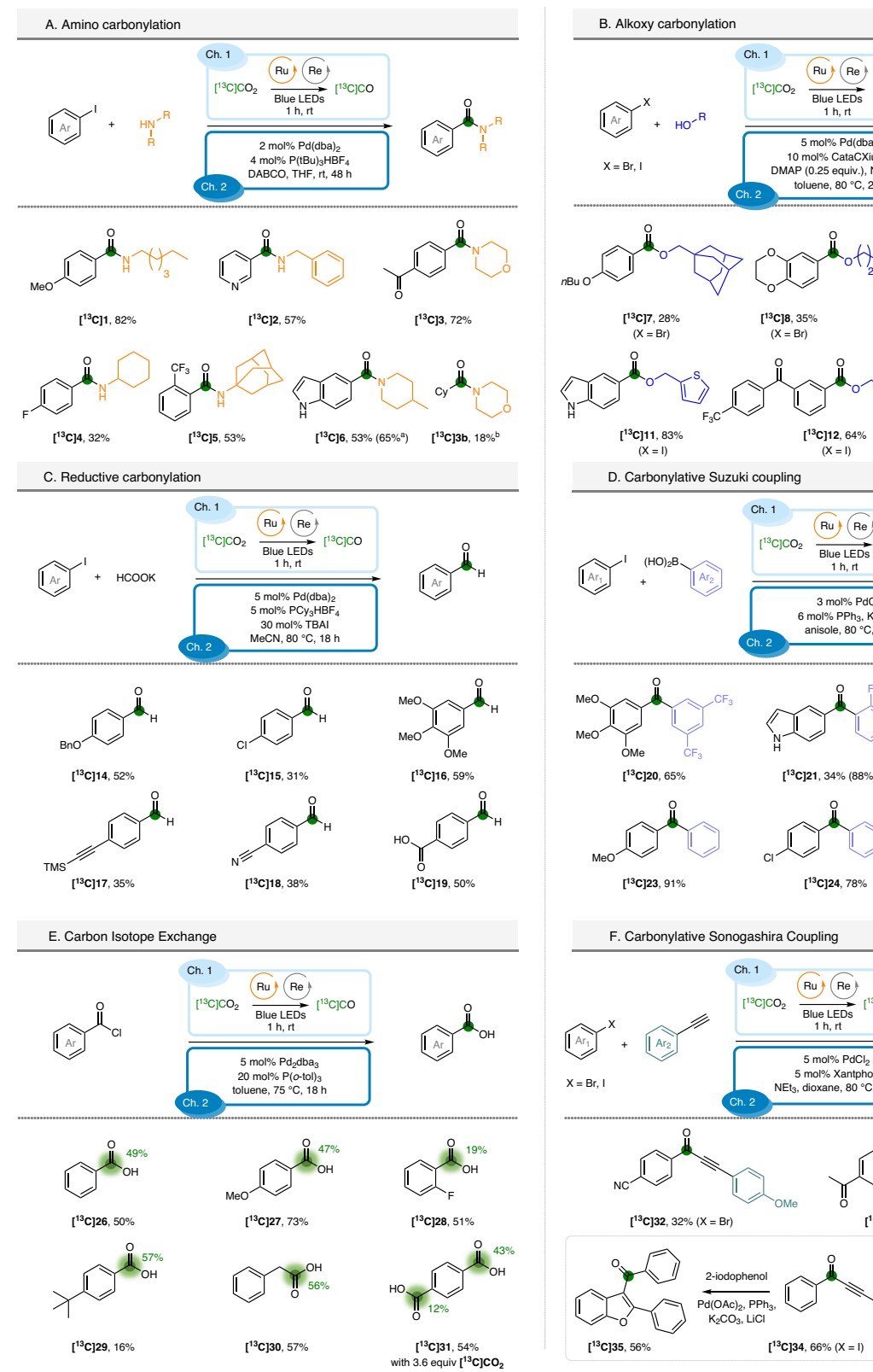

**Fig. 4 | Investigation of the scope of the photocatalytic CO₂-to-CO conversion.** Green colored circles and numbers denote the positions of the carbon atoms labeled and the percent incorporation of the carbon isotope. See Supplementary pages 40–45. Conditions in Ch1, for **A**, **B**, **C**, **E**, **F**: [¹³C]CO₂ (0.36 mmol), BIH (0.87 mmol, 2.4 equiv.); phenol (2.0 mmol, 5.5 equiv.); Ru PS (0.45 mol%, 1.62 μmol); Re(CO)₃(bpy)Br Re cat (0.45 mol%, 1.62 μmol), phosphine P2 (0.036 mmol, 0.1 equiv.) in ACN. Conditions in Ch1, for **D**: [¹³C]CO₂ (0.44 mmol), BIH (0.95 mmol, 2.12

equiv.); phenol (2.44 mmol, 5.5 equiv.); Ru PS (0.45 mol%, 1.95 μmol); Re(CO)₃(bpy) Br Re cat (0.45 mol%, 1.95 μmol), phosphine P2 (0.044 mmol, 0.1 equiv.) in ACN. [a]Conditions in Ch1: [¹³C]CO₂ (0.60 mmol), BIH (1.56 mmol, 2.60 equiv.); phenol (4.0 mmol, 6.6 equiv.); Ru PS (0.54 mol%, 3.3 μmol); Re(CO)₃(bpy)Br Re cat (0.53 mol%, 3.2 μmol), phosphine P2 (0.072 mmol, 0.12 equiv.) in ACN. [b]See Supplementary page 94.

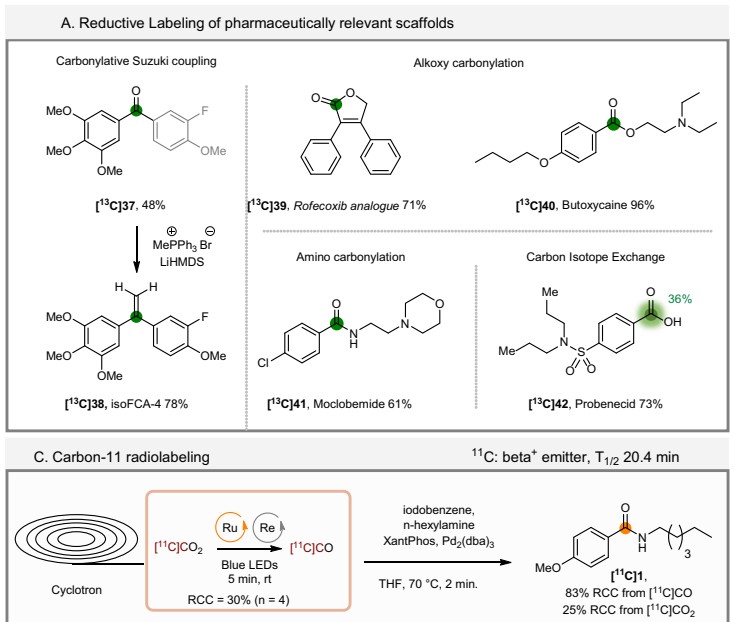

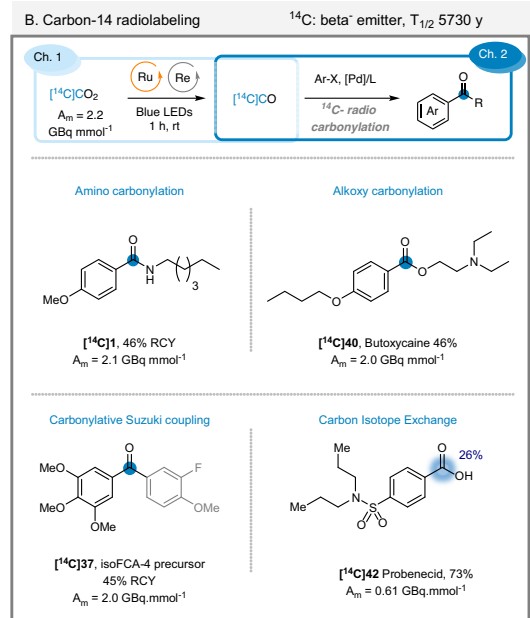

**Fig. 5 | Application of the photocatalytic $CO_2$-to-CO conversion to $^{13}$C, $^{14}$C and $^{11}$C carbonylation. A** Labeling of pharmaceutically relevant scaffolds. **B** Carbon-14 radiolabeling. **C** Carbon-11 radiolabeling. See Supplementary pages 86–109, for detailed conditions. Blue-colored circles and numbers denote the positions of the $^{14}$C atoms labeled and the percent incorporation of the isotope. Orange-colored circles denote the positions of the $^{11}$C atoms labeled.

isotopically enriched carboxylic acids (Fig. 4E). Benzoic acids **[$^{13}$C] 26–29** were isolated in moderate to good yields, in 19 to 50% isotopic enrichments (i.e., $^{13}$C/($^{13}$C+$^{12}$C) ratios). Phenyl acetic acid **[$^{13}$C]30** was also obtained in a similar degree of efficiency and finally, terephthalic acid **[$^{13}$C]31** was isolated in 43% single $^{13}$C-labeled and 12% double $^{13}$C-labeled form.

Further, we wished to implement a carbonylative Sonogashira coupling protocol (Fig. 4F). This key transformation generates synthetically useful ynones, starting from aryl halides and alkynes, whose reactivity is suitable for the construction of biologically active heterocycles. In presence of a catalytic amount of $PdCl_2$ (5 mol%) and Xantphos (5 mol%), compounds **[$^{13}$C]32–34** were isolated in 32 to 66% yield. Ynone **[$^{13}$C]34** was further converted into the benzofuran derivative **[$^{13}$C]35**, by means of an efficient *one-pot* procedure starting from *o*-iodophenol, and the pyrazole **[$^{13}$C]36**, which enabled the insertion of the carbon tag in the heterocycle core.

Having established the versatility of the photocatalytic $CO_2$-to-CO reduction over multiple carbonylative transformations (Fig. 4), we looked into the application to the labeling of biologically active molecules. **isoFCA-4 [$^{13}$C]38** (Fig. 5A), an antimitotic agent developed by the group of Alami[83], was prepared in two steps with an overall 37% yield, using a carbonylative Suzuki coupling and subsequent Wittig olefination. The Rofecoxib analogue **[$^{13}$C]39** was synthesized by an intramolecular alkoxycarbonylation in 71% yield, while the intermolecular coupling reaction allowed to isolate Butoxycaine **[$^{13}$C]40** in 96% yield and Moclobemide **[$^{13}$C]41** in 61% yield. Finally, we carried out a CIE on a Probenecid **[$^{13}$C]42** starting from the corresponding unlabeled acid with a good 36% isotopic enrichment observed.

## Radiolabeling with carbon-14 and carbon-11

To assess the importance of the photocatalytic $CO_2$-to-CO conversion in the field of radiochemistry, we next explored its application to $^{14}$C carbonylation reactions. While this step (i.e., switching from stable labeled $^{13}$C to radioactive beta emitter $^{14}$C) might look trivial, the inherent electron emission of the isotope might trigger fast covalent bond cleavage (i.e., radiolysis). The fact that [$^{14}$C]CO undergoes radiolysis is an emblematic example[28,29]. Furthermore, differences in

reactivity between stable $^{12}$C- and $^{14}$C-labeled compounds has been reported. In 1986, Parker made the hypothesis that, "if a chemical reaction can proceed by two or more pathways, where one involves the participation of free radicals, the latter might be favored, when a high molar activity radiolabeled species is involved"[84]. Consequently, the application of this photocatalytic procedure provided a high degree of uncertainty. When the standard conditions were applied to the model amide, the desired amide **[$^{14}$C]1** could be isolated in 46% radiochemical yield (RCY) and high molar activity ($A_m$) of 2.1 GBq mmol$^{-1}$. This straightforward generation of [$^{14}$C]CO was further showcased by the preparation of Butoxycaine **[$^{14}$C]40** and the isoFCA-4 precursor **[$^{14}$C]37**, both of them isolated in high $A_m$. Finally, CIE on Probenecid allowed to isolate the labeled drug **[$^{14}$C]42** in 73% yield and 37% IE ($A_m$ = 0.61 GBq mmol$^{-1}$).

At last, a proof-of-concept on $^{11}$C-photocarbonylation was realized on a Synthra module, to demonstrate the feasibility of the automation of the reaction and, therefore, the applicability to the synthesis of radiotracers for PET imaging studies. Cyclotron-produced [$^{11}$C]$CO_2$ was photochemically converted into [$^{11}$C]CO within only 5 min under mild conditions compared to the standard methods reported in the literature[34], and with a conversion of 30% ($n = 4$) comparable to the recent method described by Bongarzone et al.[36,38,39] using the disilane strategy. The separation of [$^{11}$C]CO and unreacted [$^{11}$C]$CO_2$ was easily realized by means of an Ascarite® column. Subsequent palladium-catalyzed aminocarbonylation afforded the desired compound **[$^{11}$C]1** in 83% conversion from [$^{11}$C]CO and 25% from [$^{11}$C]$CO_2$ ($n = 3$) within only 2 min. Despite the presence of "cold" [$^{12}$C]CO in the reaction mixture coming from the Re catalyst, **[$^{11}$C]1** was obtained in good molar activity of 30 GBq/μmol, as a consequence of the high-yielding process. These results demonstrate that the scope of this photoreduction–carbonylation process can be expanded to the preparation of $^{11}$C PET tracers for in vivo imaging.

In conclusion, we have reported an effective phototocatalytic approach to enable the full $CO_2$-to-CO reduction within minutes and the direct use of the produced CO in different types of carbonylation reactions. The versatility of this reaction manifold has also shown its potential in the easier and straightforward preparation of radiotracers,

which is nowadays essential in the field of human health, such as diagnosis and drug and agrochemical developments. The optimisation of this transformation has been rationalized using reaction-model-based kinetic simulations implementing photophysical and electrochemical data. The overall process has allowed the labeling of a structurally diverse library of derivatives including, amides, ester, ketones, aldehydes and carboxylic acids in one single step from $CO_2$. This technology opens up new opportunities for the direct access to [11]C- and [14]C-labeled pharmaceuticals from their primary isotopic sources [11C]$CO_2$ and [14C]$CO_2$.

## Methods

**General procedure for catalytic for the [13C]$CO_2$ photoreduction**
In a two-chamber reaction, a suspension of Ru(bpy)$_3$Cl$_2$.6H$_2$O stock solution (1.20 mL, 1.34 mM), Re(CO)$_3$(bpy)Br stock solution (0.81 mL, 1.98 mM), ACN (0.8 mL), BIH (175 mg, 0.78 mmol), phosphine P2 (20.5 mg, 0.036 mmol), phenol (188 mg, 2.0 mmol) were transferred into Chamber 1 with a Pasteur pipette. The chambers were sealed with a screwcap fitted with a Teflon®. The adaptor was then connected to the RC Tritec® system. The solution in Chamber 1 was frozen with a liquid nitrogen bath and the chambers were degassed with vacuum pump connected with RC Tritec manifold for 10 min The stopcock was closed between the two chambers. [13C]$CO_2$ (365 µmol) was then loaded into Chamber 1 using the RC Tritec® system and the stopcock was closed between Chamber 1 and the adaptor. The loaded Two-Chamber Glassware was then disconnected from the RC Tritec® system and the suspension was warmed to room temperature. Chamber 1 was placed ca. 2 cm away from a 40 W A160WE Tuna Blue Kessil® LED lamp and photo-irradiated with the lower light intensity for 1 h. The [13C]CO produced is then used in the carbonylation reaction in Chamber 2.

## Data availability

All data supporting the findings of this study are available within the article and its Supplementary Information. Details about materials and methods, experimental procedures, characterization data, and NMR spectra are available in the Supplementary Information.

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

## Acknowledgements

This work was supported by the French National Research Agency (CHARMMMAT, ANR-11-LABX-0039; fellowships to S.M.), CNRS, University Paris-Saclay the International Isotope Society—European Division (IIS-ED), the European Union's Horizon 2020 research and innovation program under the European Research Council (ERC-2019-COG—864576) and Marie Sklodowska-Curie (GA N°675071). The authors thank A. Goudet, S. Lebrequier and D.-A. Buisson (DRF-JOLIOT-SCBM, CEA) for the excellent analytical support.

## Author contributions

S.M. optimised the $CO_2$-to-CO reduction prepared the manuscript. S.M., A.T., and A.D.V. performed the experiments, synthesized and characterized the molecules, analyzed the data discussed the results. P.G., W.L. carried out the kinetic simulation. F.T. and A.A. discussed the results. O.L. and A.S. performed the carbon-14 labeling. F.C. performed the carbon-11 experiments. A.M. performed the experiments requested over the revision process. D.A. and Z.H. conceived and directed the project and prepared the manuscript.

## Competing interests

The authors declare no competing interests.
