## [Peer Review File · Nature Communications]

REVIEWER COMMENTS

Reviewer #1 (Remarks to the Author):

This manuscript by Audisio, Halime and coworkers described a novel photocatalytic reduction of CO₂ to CO, followed by applications in Pd-catalyzed carbonylation. This reaction was operated within minutes and the high efficiency enabled the facile and straightforward preparation of radiotracers, including amides, esters, acids and ketones. Although similar design has been realized by He's group and the authors' group, it is noteworthy that higher CO₂ conversion was obtained with this new methodology and applied to generate several pharmaceuticals with ¹¹C, which has not been realized in previous reactions. As well known, the synthesis of ¹¹C-labelling molecules is very challenging due to the short half-life of ¹¹C. This reviewer agreed with the opinions of the authors that this technology would open up new opportunities for direct access to ¹¹C- and ¹⁴C-labeled pharmaceuticals from their primary isotopic sources [¹¹C]CO₂ and [¹⁴C]CO₂. Besides, this work is done in details and the proposed mechanism is reasonable. Thus, the reviewer suggests this work is important and should be suitable to be published on Nat. Commun. after revisions.

1. Would increasing the pressure of CO₂ in chamber 1 affect the reaction yield in chamber 2?
2. The addition of PPh₃ seems to have little effect on the reaction, judged from the results of entries 7 and 8, table 1. Could the PPh₃ be avoided by adjusting the amount of catalyst to ensure a high yield of product?
3. This reviewer is curious about how catalytic amount of PPh₃ improved the yield of product 1 as an oxygen atom acceptor. Related references were supposed to be cited if the results of experiment failed to provide the evidence. Maybe PPh₃'s role was not abstracting the oxygen and the statement of "oxygen atom acceptor" was inaccurate.
4. Have the authors investigated other Re complexes or other Lanthanide complexes as the catalysts? Did the anion on the Re cat affect this reaction?
5. Compared with the previous work by He's group and the authors' group, what factors did the author think was the most important one to influence CO₂ to CO conversion?
6. Previous mechanistic studies were conducted according to the reactions in electrocatalytic system. While this work was conducted under photochemical conditions. There might exist difference, such as the issue of ECEC or EECC pathways caused by the distinguish reducibility of electrodes and Ru photocatalysts. The authors should conduct some experiments to address this question. Besides, DFT calculations shows that electron density of Re in Scheme 3d was located on bipyridine, suggesting that bipyridine was reduced instead of the Re center. So, this reviewer was wondering if introducing electron-withdrawing group on bipyridine in Re cat would promote this reaction to enable a shorter reaction time.
7. Was Re-H species the possible intermediate to react with CO₂? The authors were supposed to provide some evidence for this possibility.

8. In the condition, BIH was added as stoichiometric electron donor, which might also be used as proton source after oxidation process. Did the authors conduct the experiments to verify or exclude the proton source coming from BIH. For example, deuterated BIH (BID) might be used to detect D₂O or some other chemicals after being captured.

9. In this work, CO₂ was reduced by reductant to CO, followed by the carbonylation reaction. It reminds this reviewer that Yu's group has proposed a concept of "CO₂ = CO + [O]". Thus, some relative works or reviews should be cited, such as *Angew. Chem. Int. Ed.* 2016, 55, 7068; *Org. Lett.* 2018, 20, 3776; *Chem. Commun.* 2020, 56, 8355.

10. Some mistakes are present in the manuscript. Please check them carefully.

1) In SI, reference 9, "Journal of the American Chemical Society" should be "J. Am. Chem. Soc."

2) In Scheme 4, the reaction conditions in formulas seem incomplete and BIH and additive were suggested to be added.

Reviewer #2 (Remarks to the Author):

In this manuscript, Audiso and co-workers reported an effective method to enable the full CO₂-to-CO reduction via photo catalysis, which is applicable to produce CO directly in different carbonylation reactions. Considering the high efficiency of this transformation which could be achieved within 10 minutes, the mild conditions, good yields, and good applicability to different isotopically labeled CO₂, This reviewer recommends its publication in Nature Communication with following suggestions for revisions:

[1] Whether PPh₃ benefits the reaction by acting as a ligand or base? A possible conjecture that how does PPh₃ promote the reaction needs to be provided?

[2] In the proposed mechanism Scheme 3d, the double-headed arrows of reaction 8-10 may confuse readers. Although the reaction is reversible, the main pathways to achieve catalytic cycle need to be clearly drawn. And some valences of Re in the complexes are marked but some are not, they may need to be marked also.

[3] In the substrate scope part, all the substrates suitable for carbonylation reactions are aryl substrates. Whether carbonylation of alkyl or benzyl substrates can be achieved by this reaction?

[4] ¹⁹F NMR spectrum pictures of fluorine-containing products are need to be supplied in the supporting information.

Reviewer #3 (Remarks to the Author):

In their work, Halime, Audisio, and co-workers report the photochemical valorization of CO into added-value chemicals. The chemistry is based on a rapid CO₂-to-CO conversion and a followed by a valorization into radiochemicals.

The manuscripts is well organized and clearly presented. The authors have deeply investigated the reaction mechanism (Scheme 3 and SI), that is based on solid experimental evidence (pressure, temp., pKa etc.) as well as DFT calculations. The process is general and various structurally-diversified scaffolds has been examined to prove its robustness (Scheme 4). Finally, the authors have applied their process to the labeling of relevant pharmaceuticals, exploring both Carbon-11 and Carbon-14 radiolabeling.

In my opinion, this work is highly relevant to the chemical community. In particular, the developed synthetic method opens new mechanistic avenues to access ¹¹C and ¹⁴C labeled bioactive molecules. I am pleased to support its publication in Nat. Commun. after the following minor issues will be addressed.

1) The authors report the use of “blue” light. I think the emission spectra of the light-source should be reported in the main manuscript together with the absorption profile of the photocatalyst (PC). This should be also commented along the text in the optimization-of-the-process section.

2) About this part. Did the authors try to use more sustainable and abundant organic photocatalysts instead of the Ru (or Ir) metal complexes? 4CzIPN is a very versatile alternative, that is now widely use to replace several types of metal-based PCs. Other alternatives are also available (see e-g.: Chem. Commun. 2022, 58, 1263–1283).

3) Additional control experiments (that may appear superfluous) should be reported within Scheme 2a. Is the reaction working without the PC? Or without BIH? Or without light? etc. These points should be briefly commented over the text.

4) The scope of the reaction is quite general. However, I wonder if it is possible to use also aryl chlorides instead of the Br or I derivatives – being the first ones much cheaper and generally more abundant. In the case aryl chlorides are not reactive under the developed conditions, I think it will be useful to add an entry into the reaction scope showing also the limitations of the process. This it will particularly useful to the community when testing the developed process to alternative substrates.

Answers to reviewers comments

Reviewer 1:

This manuscript by Audisio, Halime and coworkers described a novel photocatalytic reduction of CO₂ to CO, followed by applications in Pd-catalyzed carbonylation. This reaction was operated within minutes and the high efficiency enabled the facile and straightforward preparation of radiotracers, including amides, esters, acids and ketones. Although similar design has been realized by He's group and the authors' group, it is noteworthy that higher CO₂ conversion was obtained with this new methodology and applied to generate several pharmaceuticals with ¹¹C, which has not been realized in previous reactions. As well known, the synthesis of ¹¹C-labelling molecules is very challenging due to the short half-life of ¹¹C.

This reviewer agreed with the opinions of the authors that this technology would open up new opportunities for direct access to ¹¹C- and ¹⁴C-labeled pharmaceuticals from their primary isotopic sources [¹¹C]CO₂ and [¹⁴C]CO₂. Besides, this work is done in details and the proposed mechanism is reasonable.

Thus, the reviewer suggests this work is important and should be suitable to be published on Nat. Commun. after revisions.

1. Would increasing the pressure of CO₂ in chamber 1 affect the reaction yield in chamber 2?

Answer: We thank the referee for the interesting question. It is indeed highly possible that the pressure increase in Chamber 1 by using excess of CO₂ (and after reduction CO), as carbonylation reactions are known to be highly sensitive to the pressure of carbon monoxide and excess of this reagent (see for ex: Acc. Chem. Res. 2014, 47, 1563–1574, Chem, 2019, 5, 526–552).

We intentionally decided not to follow that path, as our aim is primarily to provide a method that is suitable and directly implementable to radiolabelling. As such, we have to take into account considerations that are in oppositions with the use of excess CO or pressure. For example :

- The availability of radioactive ¹⁴C precursor [¹⁴C]CO₂ is limited by the high price 1Ci accounted for 25 k€ (this corresponds to ca 800 mg ¹⁴C-labeled CO₂). Consequently, use of radioactive material as limiting reagent is ideal (when possible).
- Limiting the generation of radioactive waste is a major concern, due to legislation and safety reasons. Excess of [¹⁴C]CO would immediately become a waste after the reaction. Given its instability due to radiolysis, we would not be able to store it for long-time.
- The utilisation of low-pressure reactions is a current and timely challenge in the field of long-lived radioisotope. See for example this recent article on tritium isotope labeling: *Green Chem.*, 2022, **24**, 4824-4829.
- [¹¹C]CO₂ can be generated only at the nanomole scale (due to the short half-life of 20.4 min). As such, gas pressure is not conceivable.

We have added this information in the manuscript at reference 34 (highlighted in color).

2. The addition of PPh₃ seems to have little effect on the reaction, judged from the results of entries 7 and 8, table 1. Could the PPh₃ be avoided by adjusting the amount of catalyst to ensure a high yield of product?

Answer: We have previously tried to increase the amount of catalyst only in presence of PPh₃ and by increasing it to 0.6 mol% the yield decreases to 44% (see entry 22, Table S1, reaction time 21h).

In agreement with the question of the referee, we have attempted the reaction using 0.75 mol% of Re PC and 0.75 mol% of Ru PS, at the same time avoiding the use of phosphine (entry 33, Table S1, reaction time 1h). Under such conditions, the product was obtained in 55% yield. This result is considerably lower than the standard conditions with 0.45 % Re PC and phosphine. As such, the impact of the phosphine seems to be higher on the short (1h) versus longer (21h) reactions time

A comment on these novel results has been added at page 6 of the manuscript.

3. This reviewer is curious about how catalytic amount of PPh₃ improved the yield of product 1 as an oxygen atom acceptor. Related references were supposed to be cited if the results of experiment failed to provide the evidence. Maybe PPh₃'s role was not abstracting the oxygen and the statement of "oxygen atom acceptor" was inaccurate.

Answer: PPh₃ was reported as oxygen atom acceptor in particular in oxygen atom transfer reactions (see references below). However, as described in the manuscript, the role of PPh₃ is still not clear at present, and merits further investigation. Our current efforts have only concluded the following: (1) it is not a stoichiometric oxygen acceptor, (2) it does not affect the redox potential of the Re catalyst, indicating that it might not play the role of a ligand of the catalyst (3) it does not affect the electrocatalytic activity of the Re.

Y.-M. Lee, M. Yoo, H. Yoon, X. Li, W. Nam and S. Fukuzumi Chem. Commun., 2017,53, 9352-9355

F. Avenier, C. Herrero, W. Leibl, A. Desbois, R. Guillot, J.-P. Mahy and A. Aukauloo, Angew. Chem., Int. Ed., 2013, 52, 3634.

J. Cho, J. Woo and W. Nam, J. Am. Chem. Soc., 2012, 134, 11112.

M. T. Kieber-Emmons, J. Annaraj, M. S. Seo, K. M. Van Heuvelen, T. Tosha, T. Kitagawa, T. C. Brunold, W. Nam and C. G. Riordan, J. Am. Chem. Soc., 2006, 128, 14230.

E. A. Ison, J. E. Cessarich, N. E. Travia, P. E. Fanwick and M. M. Abu-Omar, J. Am. Chem. Soc., 2007, 129, 1167.

J. P. T. Zaragoza, R. A. Baglia, M. A. Siegler and D. P. Goldberg, J. Am. Chem. Soc., 2015, 137, 6531 —6540

4. Have the authors investigated other Re complexes or other Lanthanide complexes as the catalysts? Did the anion on the Re cat affect this reaction?

Answer: Other Re or lanthanide complexes were not investigated as catalysts for the photocatalytic CO₂-to-CO reduction part because as far as modifications of the catalyst is concerned, the original Lehn's catalyst, Re bipyridine triscarbonyl bromide, was sufficiently performing. There is an array of literature in optimizing the performance of the catalyst (in conjunction with Ru as photosensitizer), and we opted to utilize the simple and effective catalyst for translation of its applicability in the two chamber system discussed herein. Of course, the authors hope that the study would open further optimizations, including the exploration of other modification on the Re catalyst or using earth abundant ones like Mn analogues.

5. Compared with the previous work by He's group and the authors' group, what factors did the author think was the most important one to influence CO₂ to CO conversion?

Answer: We believe that the most important parameter to influence the CO₂ to CO conversion are the : a) the use of phenol as proton donor and its stoichiometry; b) increase the catalyst charge (Ru and Re) from 0.15 mol% to 0.45 mol%. Here it is important to note that in presence of TEOA, 0.45 mol% was still not effective (indicating the need for phenol is probably the most important factor).

6. Previous mechanistic studies were conducted according to the reactions in electrocatalytic system. While this work was conducted under photochemical conditions. There might exist difference, such as the issue of ECEC or EECC pathways caused by the distinguish reducibility of electrodes and Ru photocatalysts. The authors should conduct some experiments to address this question. Besides, DFT calculations shows that electron density of Re in Scheme 3d was located on bipyridine, suggesting that bipyridine was reduced instead of the Re center. So, this reviewer was wondering if introducing electron-withdrawing group on bipyridine in Re cat would promote this reaction to enable a shorter reaction time.

Answer: Indeed, there might be differences between the electrocatalytic and photocatalytic systems. Both need separate investigations but the thermodynamic information of the former is quite useful and translatable to the latter. This is especially applicable in our system where Ru and Re are playing independent roles, the former as photosensitizer and the latter being the catalyst. We are using the reducing power of Ru(I) to perform electron transfers to the Re catalyst, so any information derived from DFT calculations and electrochemical simulations of the Re-cat regarding the redox potential of intermediate states are useful to build the thermodynamic landscape of the photochemical events. The much-needed trapping and tracking of catalytic intermediates are still lacking and remains an important challenge for both electrocatalytic and photocatalytic systems, meriting an intensive research on its own. As such, we relied mostly on the current knowledge on Re catalytic system based mostly on DFT and electrochemistry. Distinguishing different pathways in photocatalysis is nearly impossible because of the lack of spectral follow-up of the intermediate species, which is similarly challenging in electrocatalytic system. We opted to model the kinetics of the photocatalytic production of CO with the generally accepted pathway in Re catalytic systems, with the role of the phenol affecting the rate determining protonation step, which generally agrees with the experimental data on hand.

Indeed, it is well-known in the literature that the first reduction of Re catalyst is ligand centered (also shown in scheme 3D). There have already been numerous studies on the modifications of the bipyridine ligand and introducing electron withdrawing groups leads to a decrease in the reaction rate because of the a less nucleophilic catalyst towards the CO₂ activation (see: *ACS Catal.* **2018**, *8*, 2021–2029, *Eur JIC* **2006**, *2006*, 2966–2974). Catalyst design can be improved by a finding the right compromise between push and pull electronic effects, and by installing second coordination sphere functionalities to activate the substrate (see our reviews on catalyst design: *ChemElectroChem* **2021**, *8*, 3472–3481; *Dalton Trans.* **2020**, *49*, 2381–2396). These are currently being independently investigated in the group, and is out of the scope of the current study.

7. Was Re-H species the possible intermediate to react with CO₂? The authors were supposed to provide some evidence for this possibility.

Answer: The photocatalytic system was optimized for the production of CO, and this necessarily implies the reaction mechanism proceeds through the formation of a Re-CO₂ intermediate. We don't discard the possibility of Re-H intermediate formation prior to a reaction with CO₂, but this scenario would lead to the production of formate, which was not observed for our system. Nevertheless, past studies from the group of Kubiak (*Proc. Natl. Acad. Sci. USA* **2012**, *109*, 15646–15650) have indicated that the active form of the Re catalyst is kinetically selective towards the formation of Re-CO₂ intermediate even though Re-H intermediate is thermodynamically more stable, explaining thus the inclination of selectivity towards CO production.

8. In the condition, BIH was added as stoichiometric electron donor, which might also be used as proton source after oxidation process. Did the authors conduct the experiments to verify or exclude the proton source coming from BIH. For example, deuterated BIH (BID) might be used to detect D₂O or some other chemicals after being captured.

Answer: In our previous study (ref 15) we have concluded that BIH is a two-electron, one-proton donor. Since CO₂-to-CO requires two electrons and two protons, it was necessary to introduce an additional source of protons, i.e. phenol for the system. We have shown that with sodium phenolate, despite having BIH in the solution, the reaction does not occur. This indicates that stoichiometric amounts of H coming from BIH is not sufficient, and an second proton source (in excess) is need to push the reaction forward. As such, we believe that the proposed deuterated experiment will not give access to additional information.

9. In this work, CO₂ was reduced by reductant to CO, followed by the carbonylation reaction. It reminds this reviewer that Yu's group has proposed a concept of "CO₂ = CO + [O]". Thus, some relative works or reviews should be cited, such as Angew. Chem. Int. Ed. 2016, 55, 7068; Org. Lett. 2018, 20, 3776; Chem. Commun. 2020, 56, 8355.

Answer: We thank the referee for the suggestion. The references have been cites in the introduction of the article (ref 7-9).

10. Some mistakes are present in the manuscript. Please check them carefully.

1) In SI, reference 9, "Journal of the American Chemical Society" should be "J. Am. Chem. Soc."

Answer: The correction has been implemented in the supporting information. All the reference in the SI have been homogenized as well.

2) In Scheme 4, the reaction conditions in formulas seem incomplete and BIH and additive were suggested to be added.

Answer: We agree with the comment. In terms of space it is difficult to insert the precise reactions conditions in Scheme 4. Nonetheless, we have added the experimental details for Ch1 in the caption of scheme 4.

Reviewer 2:

In this manuscript, Audisio and co-workers reported an effective method to enable the full CO₂-to-CO reduction via photo catalysis, which is applicable to produce CO directly in different carbonylation reactions. Considering the high efficiency of this transformation which could be achieved within 10 minutes, the mild conditions, good yields, and good applicability to different isotopically labeled CO₂.

This reviewer recommends its publication in Nature Communication with following suggestions for revisions:

[1] Whether PPh₃ benefits the reaction by acting as a ligand or base? A possible conjecture that how does PPh₃ promote the reaction needs to be provided?

Answer: We have admitted in the manuscript that the role of PPh₃ is still not clear, and merits further investigation. Our current efforts have only concluded the following: (1) it is not a stoichiometric oxygen

acceptor, (2) it does not affect the redox potential of the Re catalyst, indicating that it might not play the role of a ligand of the catalyst (3) it does not affect the electrocatalytic activity of the Re.

[2] In the proposed mechanism Scheme 3d, the double-headed arrows of reaction 8-10 may confuse readers. Although the reaction is reversible, the main pathways to achieve catalytic cycle need to be clearly drawn. And some valences of Re in the complexes are marked but some are not, they may need to be marked also.

Answer: We thank the reviewer for this suggestion. The suggestion is now shown in the revised version of the manuscript.

The oxidation state of the Re was explicitly shown in our initially submitted manuscript for the first three species, because these are experimentally confirmed; but the oxidation states of Re within the catalytic cycle is not indicated because of lack of experimental evidence and the dynamic and often mixed metal-ligand character during catalysis. To maintain standard and uniform notations in the catalytic cycle, we opted now to not show the oxidation state of the metal center. The electron and proton transfers are indicated and can be followed in the overall charge of the intermediate species, indicated in the upper right brackets. The double-headed arrows are now avoided for clarity. In addition, thicker lines are indicated for the system that uses water as proton source while the green arrows indicate the changes once phenol is used.

[3] In the substrate scope part, all the substrates suitable for carbonylation reactions are aryl substrates. Whether carbonylation of alkyl or benzyl substrates can be achieved by this reaction?

Answer: In agreement with the suggestion, we have attempted an example of palladium-catalysed aminocarbonylation between cyclohexyl iodide and morpholine (according to the following report: *J. Org. Chem.* **2019**, *84*, 16076–16085). Without optimisation, the formation of the corresponding labeled amide was observed in the reaction crude, in 18% IS yield. This result, which was not optimised further, show that the carbonylation of alkyl substrates is possible, as well.

The experimental procedure was added in the supporting information at page S91 and ref 71 was added in the manuscript to report this experiment.

[4] ¹⁹F NMR spectrum pictures of fluorine-containing products are need to be supplied in the supporting information.

Answer: We apologize about that. The pictures of the ¹⁹F-NMR of the nine fluorine-containing products have now been added in the SI.

Reviewer 3:

In their work, Halime, Audisio, and co-workers report the photochemical valorization of CO into added-value chemicals. The chemistry is based on a rapid CO₂-to-CO conversion and a followed by a valorization into radiochemicals.

The manuscripts is well organized and clearly presented. The authors have deeply investigated the reaction mechanism (Scheme 3 and SI), that is based on solid experimental evidence (pressure, temp., pKa etc.) as well as DFT calculations. The process is general and various structurally-diversified scaffolds has been examined to prove its robustness (Scheme 4). Finally, the authors have applied their process to the labeling of relevant pharmaceuticals, exploring both Carbon-11 and Carbon-14 radiolabeling.

In my opinion, this work is highly relevant to the chemical community. In particular, the developed synthetic method opens new mechanistic avenues to access ^{11}C and ^{14}C labeled bioactive molecules. I am pleased to support its publication in Nat. Commun. after the following minor issues will be addressed.

1) The authors report the use of “blue” light. I think the emission spectra of the light-source should be reported in the main manuscript together with the absorption profile of the photocatalyst (PC). This should be also commented along the text in the optimization-of-the-process section.

Answer: The normalized emission spectra of the light sources (Kessil and SugarCube blue lamps) and the absorption profile of the Ru PS have now been added to the Supplementary Information: Figure S26.

The comments have been integrated in the manuscript, as well, at page 5 and reference 34 (modifications highlighted with color).

Fig. S26. Normalized emission spectra of the Kessil (red) and SugarCube (blue, used in previous study, see: *ChemPhotoChem* **2018**, 2, 715) LED lamps, as well as the molar absorptivity profile of the ruthenium photosensitizer (black).

2) About this part. Did the authors try to use more sustainable and abundant organic photocatalysts instead of the Ru (or Ir) metal complexes? 4CzIPN is a very versatile alternative, that is now widely used to replace several types of metal-based PCs. Other alternatives are also available (see e.g.: *Chem. Commun.* 2022, 58, 1263–1283).

Answer: There is indeed a trending field in replacing Ru photosensitizers with organic dyes. The current study was limited to the Ru photosensitizer because its photochemistry is well-established allowing an eraser monitoring of the spectral changes in our photophysics investigations (transient absorptions). Replacing it by other photosensitizers, especially the organic ones, would require additional systems optimization (identification of the proper light source, determination of electron transfer kinetics to Re catalyst, etc.) which is out of the scope of this study. We hope however that this study would instigate further optimizations in such lines as well as going toward earth abundant catalysts.

We have tried preliminary experiments with 4CzIPN, as indicated by the referee and these entries have been added in the SI in the optimization table S1 (entries 35-37). When CZIPN replaced the [Re ca] and [Ru PS] no product was formed (<3%). When 4CzIPN was used in presence of [Re Cat] we could observe 40% amide formation. This result is the same as the background reaction in the absence of [Ru PS]. Overall, these data support that the organic PC 4CzIPN is not suitable in the CO₂-to-CO conversion.

Nonetheless, we cannot exclude that other organic photocatalysts with reducing potentials might be suitable for the transformation.

We have added a short comment on this at reference 34 in the manuscript.

3) Additional control experiments (that may appear superfluous) should be reported within Scheme 2a. Is the reaction working without the PC? Or without BIH? Or without light? etc. These points should be briefly commented over the text.

Answer: We appreciate the comment. Part of these control experiments were already been reported in the SI on the 21h reaction (see Table S1, entries 10, 11, 12).

In agreement with the comment of referee 3, we have performed additional control experiments on 1h optimised conditions and added it in Table S1. When performed without BIH (entry 29), Re Cat (entry 30) or light (entry 34) no desired product was observed. Without phenol only 10% IS yield was observed (entry 31), while in the absence of Ru PS a 42% yield was obtained (entry 28).

We have added a brief comment in the text (highlighted with color).

4) The scope of the reaction is quite general. However, I wonder if it is possible to use also aryl chlorides instead of the Br or I derivatives – being the first ones much cheaper and generally more abundant. In the case aryl chlorides are not reactive under the developed conditions, I think it will be useful to add an entry into the reaction scope showing also the limitations of the process. This it will particularly useful to the community when testing the developed process to alternative substrates.

Answer: We thank the referee for the useful question. From our initial scope, we indeed have noticed that aryl chloride substitutions are tolerated in presence of aryl iodides, which react selectively (see [¹³C]15 and [¹³C]24. These results seem to indicate that the ArCl are ineffective substrates.

In accordance with the suggestion by the referee, the following aryl chlorides were used in the amino- and Suzuki-carbonylations. We expressly selected reaction partners that worked effectively from the identical Ar-I derivatives. Under otherwise identical reaction conditions, no formation labeled products was observed.

(a) Conditions : GP1

(b) Conditions : GP3

We have added this reaction in the SI part of the article and a comment has been included in the manuscript, as well.

REVIEWERS' COMMENTS

Reviewer #1 (Remarks to the Author):

Audisio, Halime and coworkers described a novel photocatalytic CO₂ reduction to CO, followed by applications in Pd-catalyzed carbonylation. It is significant and contributes to CO₂ utilization and (radio)labeling community. The authors have answered most of my questions, thus I support the publication of this article. Besides, I have several suggestions for authors:

1. For the answers to question 1 by Reviewer 1, I understand the authors' intention to avoid the use of excess CO or pressure in the current method. Those opposite considerations proposed by the authors are reasonable. My opinion is that this work is mainly a methodological study, which is successful in the laboratory. Thus, the authors were suggested to provide as much information as possible for further practical application research. Especially in the case of substrates that delivered the products in moderate yields, increasing the pressure of CO₂ might be useful to improve the reaction yield in chamber 2.
2. For the answers to question 5 by Reviewer 1, the authors referred to several factors to influence CO₂ to CO conversion. Which one was the most determining one? How about the type of catalyst? Besides, these conclusions were supposed to be discussed in the main text.
3. Reaction formula of those experiments conducted to respond questions should be added in the Response Letter to make it easy for reviewers to understand.

Reviewer #2 (Remarks to the Author):

The authors carefully revised the article and seriously answered the comments. Detailed experiments and explanation were supplemented. After revision, this manuscript is more complete and rigorous. This reviewer recommends it to publish in Nature Communication. The supplementary results that the carbonylation of alkyl substrates is possible could be added in the article to complete the substrate range. Scheme 2 and Scheme 3 in the PDF were fuzzy, clearer pictures may be needed.

Reviewer #3 (Remarks to the Author):

Halime, Audisio and co-workers have carefully addressed all the issues raised from this reviewer. I thus support the acceptance of this work as it stands.

Answers to reviewers comments: 2nd Revision

Reviewer #1 (Remarks to the Author):

Audisio, Halime and coworkers described a novel photocatalytic CO₂ reduction to CO, followed by applications in Pd-catalyzed carbonylation. It is significant and contributes to CO₂ utilization and (radio)labeling community. The authors have answered most of my questions, thus I support the publication of this article. Besides, I have several suggestions for authors:

1. For the answers to question 1 by Reviewer 1, I understand the authors' intention to avoid the use of excess CO or pressure in the current method. Those opposite considerations proposed by the authors are reasonable. My opinion is that this work is mainly a methodological study, which is successful in the laboratory. Thus, the authors were suggested to provide as much information as possible for further practical application research. Especially in the case of substrates that delivered the products in moderate yields, increasing the pressure of CO₂ might be useful to improve the reaction yield in chamber 2.

Answer: In agreement to the suggestion by the referee, we have performed two reactions on substrates that have been isolated with moderate yields under standard procedures.

We have selected amide [¹³C]6 and ketone [¹³C]21, that were previously isolated in 53% and 34% yield.

The reactions were performed using 600 μmol of [¹³C]CO₂ in chamber 1 (catalyst loading was increased accordingly), while in chamber B (carbonylative reaction) the conditions were identical. As expected, under higher stoichiometry of CO the yields were increased in both case to 65% and 88%.

This information was added in Scheme 4 and in the supporting information, as well. Two sentences were added in the main text (highlighted with colour).

2. For the answers to question 5 by Reviewer 1, the authors referred to several factors to influence CO₂ to CO conversion. Which one was the most determining one? How about the type of catalyst? Besides, these conclusions were supposed to be discussed in the main text.

Answer: We did not screen many type of catalysts. We essentially focused on [Re cat], but we have highlighted that organic photocatalyst 4CzIPN was not effective. As previously answered, the proton donor is in our opinion the most effective factor to improve CO₂ to CO conversion.

As suggested, we have added the discussion in the main text (highlighted with colours).

3. Reaction formula of those experiments conducted to respond questions should be added in the Response Letter to make it easy for reviewers to understand.

Answer: We apologize for that. We have highlighted in the SI, previously.

In this second reply, we have included formulas when additional experimentation was required.

Reviewer #2 (Remarks to the Author):

The authors carefully revised the article and seriously answered the comments. Detailed experiments and explanation were supplemented. After revision, this manuscript is more complete and rigorous. This reviewer recommends it to publish in Nature Communication.

1) The supplementary results that the carbonylation of alkyl substrates is possible could be added in the article to complete the substrate range.

Answer: We have added this information in Scheme 4 (cpd [13C]3b) and a sentence in the main text (highlighted with colour), as well.

2) Scheme 2 and Scheme 3 in the PDF were fuzzy, clearer pictures may be needed.

Answer: We tried to increase the quality of the pictures.

Reviewer #3 (Remarks to the Author):

Halime, Audisio and co-workers have carefully addressed all the issues raised from this reviewer. I thus support the acceptance of this work as it stands.

Answer: We thank reviewer 3 for his appreciation, and help to improve the quality of our manuscript.